# A CAUSAL VIEW ON
# ROBUSTNESS OF NEURAL NETWORKS

## ABSTRACT

We present a causal view on the robustness of neural networks against input manipulations, which applies not only to traditional classification tasks but also to general measurement data. Based on this view, we design a deep causal manipulation augmented model (deep CAMA) which explicitly models the manipulations of data as a cause to the observed effect variables. We further develop data augmentation and test-time fine-tuning methods to improve deep CAMA's robustness. When compared with discriminative deep neural networks, our proposed model shows superior robustness against unseen manipulations. As a by-product, our model achieves disentangled representation which separates the representation of manipulations from those of other latent causes.

## 1 INTRODUCTION

Deep neural networks (DNNs) have great success in many real-life applications, however, they are easily fooled even by a tiny amount of perturbation (Szegedy et al., 2013; Goodfellow et al., 2015; Carlini & Wagner, 2017b; Athalye et al., 2018). Lack of robustness hinders the application of DNNs to critical decision making tasks such as uses in health care. To address this, a deep learning practitioner may suggest training DNNs with datasets that are not only big but also diverse. Indeed, data augmentation and adversarial training have shown improvements in both the generalization and robustness of DNNs (Kurakin et al., 2016; Perez & Wang, 2017; Madry et al., 2017). Unfortunately, this does not address the vulnerability of DNNs for unseen manipulations. For example, as shown in Figure 1, a DNN trained on clean MNIST digits fails to classify shifted digits. Although observing (adversarial) perturbations of clean data in training improves robustness against that particular manipulation (the green line), the DNN is still fragile when unseen manipulations are present (orange line). Since it is unrealistic to augment the training data towards all possible manipulations that many occur, a principled method that fundamentally improves the robustness is much needed.

On the other hand, humans naturally understand the independent *causal* mechanisms for visual recognition tasks, where the generative process of the perceived view is composed of modules that do not influence each other (Parascandolo et al., 2017). After learning the concept of an "elephant", a child can identify the elephant in a photo taken under any lightning condition, location, etc. Importantly, the elephant, the lightning condition, and the location are causes of the presented view in the photo. Therefore we argue that the incapability for *causal reasoning* (Pearl & Mackenzie, 2018; Gopnik et al., 2004) is the reason of DNN's vulnerability to (adversarial) data manipulations.

This work discusses the robustness of DNNs from a causal perspective. Our contributions are:

- *A causal view on robustness of neural networks.* We argue from a causal perspective that adversarial examples for a model can be generated by manipulations on the effect variables and/or their unseen causes. Therefore DNN's vulnerability to adversarial attacks is due to the lack of causal understanding.
- *A causal inspired deep generative model.* We design a causal deep generative model which takes into account the unseen manipulations of the effect variables. Accompanied with this model is a test-time inference method to learn unseen manipulations and thus improve classification accuracy on noisy inputs. Data augmentation techniques can also be safely applied to our model during training without deteriorating its generalization ability to unseen manipulations. Compared to DNNs, experiments on both MNIST and a measurement-based dataset show that our model is significantly more robustness to unseen manipulations.

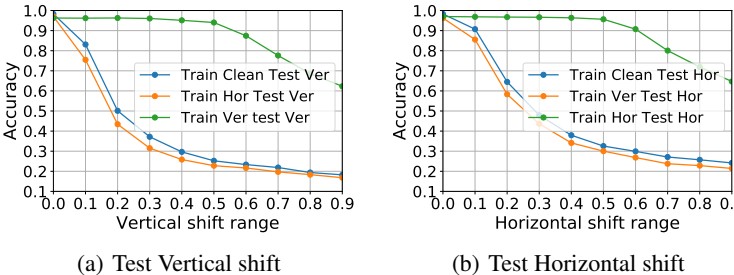

(a) Test Vertical shift          (b) Test Horizontal shift

Figure 1: Robustness results for DNNs against different manipulations on MNIST. Panels (a) and (b) show the accuracy on classifying noisy test data generated by shifting the digits vertically (Ver) and horizontally (Hor). It shows that data augmentation during training makes generalization to unseen shifts worse (orange versus blue lines).

## 2    A CAUSAL VIEW ON ROBUSTNESS OF NEURAL NETWORKS

Discriminative DNNs are not robust to manipulations such as adversarial noise injection (Goodfellow et al., 2015; Carlini & Wagner, 2017a; Athalye et al., 2018), rotation and shift. They do not understand the *causal mechanisms* of the data generating process, which leads to overfiting to nuisance factors that are less related to the ground truth classification results. By exploiting the overfit to the nuisance factors, an adversary can easily manipulate the inputs to fool discriminative DNNs into predicting the wrong outcomes.

On the contrary, we as human can easily recognize an object in a scene and be indifferent to the changes in other aspects such as background, viewing angle, the presence of a sticker to the object, etc. More importantly, our recognition is not affected even when some of the perturbations, e.g. changes in the lighting condition, are significant. We argue that the main difference here is due to our ability to perform *causal reasoning*, which identifies *independent mechanisms* that are not causally related to the object recognition results (Freeman, 1994; Peters et al., 2017; Parascandolo et al., 2017). This leads to robust human perception to not only a certain type of perturbations, but also to many types of manipulations. Thus we argue that one should incorporate causal mechanisms into model design, and make the model robust on the level of different types of perturbations.

Before presenting our causally informed model, we first define a valid manipulation of inputs in a causal sense. A valid manipulation is a perturbation on data, which only changes the effects, not the cause of the target. We visualize a causal graph in Figure 2, where the arrows indicate the cause-effect relationship between variables. Take hand-written digit classification for example, $X$ is the image of a digit and $Y$ is the class label. The appearance of $X$ is an effect of the digit number $Y$, latent causes $Z$ such as writing styles, and possible manipulations $M$, such as rotation or translation. Changes to $Z$ and $M$ cause the appearance of $X$ to change, but $X$ still carries the same information about $Y$ regardless of these perturbations, since $Z$, $M$ and $Y$ are independent mechanisms. Thus, any manipulation that does not influence the $Y \to X$ relationship are valid manipulations. Humans are extremely robust to these manipulations while machine learning algorithms are vulnerable.

In summary, from the causal perspective, any manipulation $M$ on data $X$, that is a co-parent of $Y$, is a valid manipulation. This definition includes many manipulations used in existing work on the robustness of neural networks, such as noise injection, shift and rotation (Engstrom et al., 2019). Ideally, a machine learning model should be able to generalize to any valid manipulation, at the same time training with manipulated data of certain types should never harm the model's robustness to unseen manipulations. However, discriminative deep learning models ignore the causal structure and consider $X \to Y$ only, which explains their vulnerability to data manipulations. Inspired by causal reasoning of humans, we propose a deep learning framework concerning the causal relationship.

## 3    THE CAUSAL MANIPULATION AUGMENTED MODEL

We propose a deep CAusal Manipulation Augmented model (deep CAMA), which takes into account the causal relationship for model design. Our proposed model is more robust to unseen manipulations on effect variables, and more importantly, our model can learn these manipulations

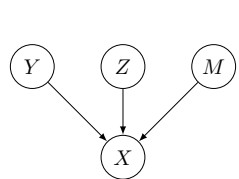

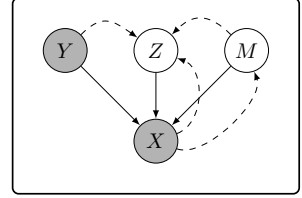

Figure 2: A simple example, where X is the effect of Y, Z and M.

Figure 3: Graphical presentation of proposed causally consistent deep generative model for single modal data.

Figure 4: The network architecture. Shaded areas show the selective part for $do(m)$ training and the fine-tune method, respectively.

without supervision. The robustness can be further improved by training-time data augmentation, without sacrificing the generalization ability to unseen manipulations. Below we first present the deep CAMA for single modality data, which focuses on predicting $Y$ using $X$, and then present a generic deep CAMA for multimodality measurement data.

### 3.1 DEEP CAMA FOR SINGLE MODALITY DATA

The task of predicting $Y$ from $X$ covers a wide range of applications such as image/speech recognition and sentiment analysis. Normally a discriminative DNN takes $X$ as input and directly predicts (the distribution of) the target variable $Y$. Generative classifiers, on the other hand, build a generative model $Y \to X$, and use Bayes' rule for predicting $Y$ given $X$: $p(y|x) = p(y)p(x|y)/p(x)$.

We design deep CAMA (Figure 3) following the causal relationship as shown in Figure 2. Taking MNIST for example: $Y$ is the label and $X$ is the image, $Z$ models the latent style of the digits, and $M$ handles the manipulations that we desire the model to be robust to. The model is defined as:

$$p_\theta(x, y, z, m) = p(m)p(z)p(y)p_\theta(x|y, z, m) \tag{1}$$

For efficient inference we follow the *amortized inference* approach in variational auto-encoders (Kingma & Welling, 2013; Rezende et al., 2014; Zhang et al., 2018) and define an inference network as the approximate posterior distribution:

$$q_\phi(z, m|x, y) = q_{\phi_1}(z|x, y, m)q_{\phi_2}(m|x). \tag{2}$$

We use $\phi$ to denote all the parameters of the encoder network and $\phi = \{\phi_1, \phi_2\}$, where $\phi_1$ is the parameter for the encoder network for the variational distribution $q_{\phi_1}(z|x, y, m)$, and $\phi_2$ is used for the $q_{\phi_2}(m|x)$ part. Note that we assume the dependence of $M$ on $X$ only in $q_{\phi_2}(m|x)$, which, as we shall show later, allows deep CAMA to learn unseen manipulations with unlabelled noisy data.

The network architecture is presented in Figure 4. For the $p$ model, the cause variables $Y$, $Z$ and $M$ are first transformed into feature vectors $h_Y, h_Z$ and $h_M$. Later, these features are merged together and then passed through another neural network to produce the distributional parameters of $p_\theta(x|y, z, m)$. For the approximate posterior $q$, two different networks are used to compute the distributional parameters of $q_{\phi_2}(m|x)$ and $q_{\phi_1}(z|x, y, m)$, respectively.

**Model training** Assume that during training, the model observes clean data $\mathcal{D} = \{(x_n, y_n)\}$ only. In this case we set the manipulation variable $M$ to a null value, e.g. $do(m = 0)$, and train deep CAMA by maximizing the likelihood function $\log p(x, y|do(m = 0))$ under training data. Since this marginal distribution is intractable, we instead maximize the *intervention* evidence lower-bound (ELBO) with $do(m = 0)$, i.e. $\max_{\theta,\phi} \mathbb{E}_\mathcal{D}[\text{ELBO}(x, y, do(m = 0))]$, with the ELBO defined as

$$\begin{aligned}
\text{ELBO}(x, y, do(m = 0)) :=& \mathbb{E}_{q_\phi(z|x,y,do(m=0))} \left[ \log \frac{p_\theta(x, y, z|do(m = 0))}{q_\phi(z, |x, y, do(m = 0))} \right] \\
=& \mathbb{E}_{q_{\phi_1}(z|x,y,m=0)} \left[ \log \frac{p_\theta(x|y, z, m = 0)p(y)p(z)}{q_{\phi_1}(z|x, y, m = 0)} \right].
\end{aligned} \tag{3}$$

See appendix A for a detailed derivation. If noisy data $\mathcal{D}'$ is available during training, then similar to data augmentation and adversarial training (Goodfellow et al., 2015; Tramèr et al., 2018; Madry et al., 2017), we can augment the training data with this noisy data. We still use the intervention ELBO (3) for clean data. For the manipulated instances, we can either use the intervention ELBO with $do(m = m_0)$ when the noisy data $\mathcal{D}' = \{(m_0(x), y)\}$ is generated by a known manipulation $m_0$, or, as done in our experiments, infer the latent variable $M$ for unknown manipulations. This is achieved by maximizing the ELBO on the joint distribution $\log p(x, y)$ using noisy data:

$$\text{ELBO}(x, y) := \mathbb{E}_{q_\phi(z,m|x,y)} \left[ \log \frac{p_\theta(x, y, z, m)}{q_\phi(z, m|x, y)} \right], \tag{4}$$

and therefore the total loss function to be maximized is defined as

$$\mathcal{L}_{\text{aug}}(\theta, \phi) = \lambda \mathbb{E}_{\mathcal{D}}[\text{ELBO}(x, y, do(m = 0))] + (1 - \lambda) \mathbb{E}_{\mathcal{D}'}[\text{ELBO}(x, y)]. \tag{5}$$

Our causally consistent model effectively disentangles the latent representation: $Z$ models the unknown causes in the clean data, such as personal writing style; and $M$ models possible manipulations which the model should be robust to, such as shift, rotation, noise etc. Due to independent mechanism assumptions in causality, the influence of $Y$, $Z$ and $M$ on $X$ can be independently applied. Thus, with our model design, we can also ensure that the dependencies $Y \to X$ and $Z \to X$ are not affected by noisy data present during training. As a result, deep CAMA's can still generalize to unseen manipulations even after seeing lots of noisy datapoints from other manipulations, in contrast to the behavior of discriminative DNNs as shown in Figure 1.

**Prediction**   In general the test data $\tilde{\mathcal{D}}$ can be noisy, and we would like our model to be robust to the unseen manipulated test data. Thus, at test-time, $M$ is unknown, and deep CAMA classifies an unseen test data $x^*$, using a Monte Carlo approximation to Bayes' rule with samples $m^u \sim q_{\phi_2}(m|x)$, $z_c^k \sim q_{\phi_1}(z|x^*, y_c, m^u)$:

$$p(y^*|x^*) = \frac{p(x^*|y^*)p(y^*)}{p(x^*)} \approx \text{softmax}_{c=1}^{C} \left[ \log \sum_{k=1}^{K} \frac{p_\theta(x|y, z_c^k, m^u)p(y_c)p(z)}{q_{\phi_1}(z_c^k|x^*, y_c, m^u)} \right]. \tag{6}$$

In addition, deep CAMA can be adapted to the unseen manipulations present at test time *without labels on the noisy data*. From the causal graph, the conditional distributions $p(X|Y)$ and $p(X|Z)$ are invariant to the interventions on $X$ based on the independent mechanism assumption (Peters et al., 2017), however, we would like to learn the manipulation mechanism $M \to X$. As shown in Figure 4, for the generative model, we only fine-tune the networks that are dependent only on $M$, i.e. $\text{NN}_M^p$ by maximizing the ELBO of the marginal distribution $\log p(x)$:

$$\text{ELBO}(x) := \log \left[ \sum_{c=1}^{C} \exp[\text{ELBO}(x, y_c)] \right]. \tag{7}$$

To reduce the possibly negative effect of fine-tuning to model generalization, we use a shallow network for $\text{NN}_{merge}^p$ and deep networks for $\text{NN}_M^p$, $\text{NN}_Y^p$ and $\text{NN}_Z^p$. We also fine-tune the network $\text{NN}_M^q$ for the approximate posterior $q$ since $M$ is involved in the inference of $Z$. In sum, in fine-tuning the selective part of the deep CAMA model is trained to maximize the following objective:[1]

$$\mathcal{L}_{\text{ft}}(\theta, \phi) = \alpha \mathbb{E}_{\mathcal{D}}[\text{ELBO}(x, y)] + (1 - \alpha) \mathbb{E}_{\tilde{\mathcal{D}}}[\text{ELBO}(x)]. \tag{8}$$

Notice that there may exist infinitely many manipulations and it is impossible to observe all of them at training time. Therefore by fine-tuning at test-time, the model can be adapted to any unseen manipulation which is desirable in many real-life applications. As shown in our experiments, the proposed deep CAMA model and the training methods are capable of improving the robustness of the generative classifier to unseen manipulations.

## 3.2   DEEP CAMA FOR GENERIC MEASUREMENT DATA

We now discuss an even more general version of deep CAMA to handle multimodality in measurement data. To predict the target variable $Y$ in a directed acyclic graph, only variables in the Markov

---

[1]One can also use the intervention ELBO for the clean training data.

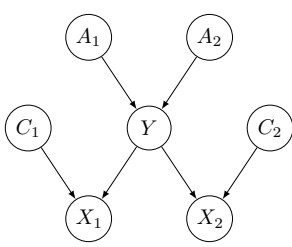

Figure 5: The Markov Blanket of target variable $Y$

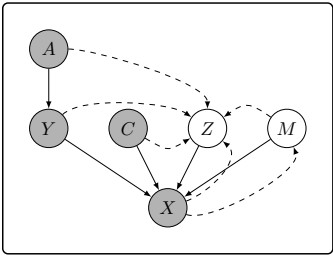

Figure 6: Graphical presentation of proposed causal deep generative model for generic measurement modal data.

blanket of $Y$ (shown in Figure 5) are needed. This includes the parents ($A$), children ($X$), and co-parents ($C$) of the target $Y$. Similar to the single modal case above, here a valid manipulation can only be independent mechanisms applied to $X$ or $C$ to ensure that $Y$ does not change and the relationship from $Y$ to $X$ does not change.

We design the generic deep CAMA (shown in Figure 6) following the causal process in Figure 5. Unlike discriminative DNNs where $A$, $C$ and $X$ are used together to predict $Y$ directly, we consider the full causal process and treat them separately. Building on the deep CAMA for single modality data, we add the extra consideration of the parent and observed co-parent of $Y$, while modelling the latent unobserved cause in $Z$ and potential manipulations in $M$. We do not need to model manipulation on $C$ as they are out of the Markov Blanket of $Y$. Thus, our model and the approximate inference network are defined as:

$$p_\theta(x, y, z, m, a, c) = p(a)p(m)p(z)p(c)p_{\theta_1}(y|a)p_{\theta_2}(x|y, c, z, m), \qquad (9)$$

$$q_\phi(z, m|x, y, a, c) = q_{\phi_1}(z|x, y, m, a, c)q_{\phi_2}(m|x). \qquad (10)$$

Training, fine-tuning and prediction proceed in the same way as in the single modality deep CAMA (Section 3.1) with $do(m)$ operations and Monte Carlo approximations. As we only fine-tune the networks that are dependent on $M$, using similar reasoning one can show that the multimodality deep CAMA is robust to manipulations directly on the effect variable $X$.

Our proposed model is also robust to manipulations on the co-parents $C$ by design. By our definition of valid manipulation, perturbing $C$ is valid as only causes the changes in $X$. If the underlying causal relationship between $C$ and $X$ remains the same, and the trained model learns $p(x|y, c)$ *perfectly*, then our model is perfectly robust to such changes. This is because we use Bayes' rule for prediction:

$$p(y|a, x, c) = \frac{p(y|a)p(a)p(c)p(x|y, c)}{p(a)p(c) \int_y p(y|a)p(x|y, c)} = \frac{p(y|a)p(x|y, c)}{\int_y p(y|a)p(x|y, c)}, \qquad (11)$$

and the manipulations on $C$ (thus changing $X$) do not affect the conditional distribution $p(x|y, c)$ in the generative classifier (Eq. 11). In contrast, discriminative DNNs concatenate $X$, $C$, $A$ together and map these variables to $Y$, therefore they are sensitive to manipulations on $C$ and/or $X$.

## 4 EXPERIMENTS

In this section, we first show the robustness of our proposed deep CAMA for image classification using both MNIST and a binary classification task derived from CIFAR-10. Then, we demonstrate the behaviour of our generic deep CAMA for measurement data. We evaluated the perfromance of CAMA on both manipulations such as shifting and adverserial examples generated using the CleverHans package (Papernot et al., 2018). More results with different DNN architectures and different manipulations are shown in the appendix.

### 4.1 ROBUSTNESS TEST ON IMAGE CLASSIFICATION WITH DEEP CAMA

We first demonstrate the robustness of our model against vertical (VT) and horizontal (HT) shifts. Details such as network architectures are presented in the appendix. The experiments are repeated for 5 times, and on MNIST, the results are stable and the variances are not visible in the plot.

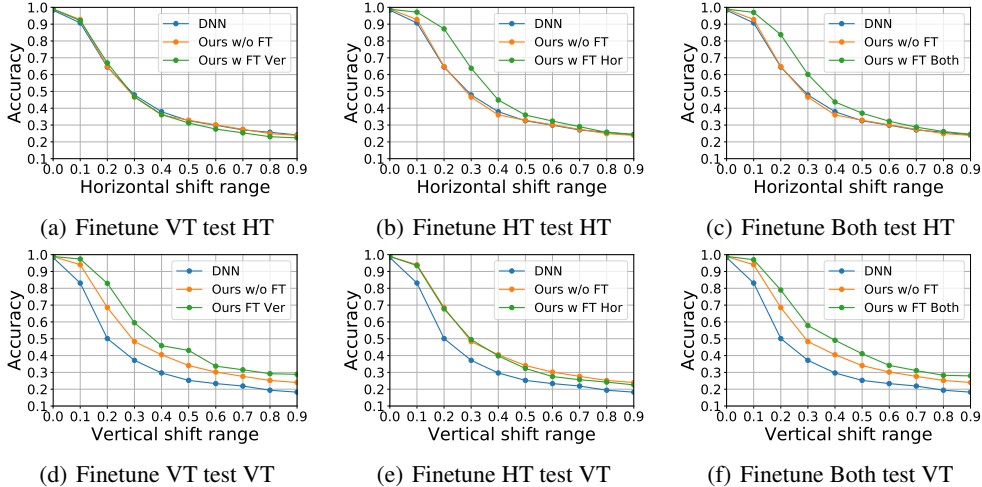

(a) Finetune VT test HT     (b) Finetune HT test HT     (c) Finetune Both test HT

(d) Finetune VT test VT     (e) Finetune HT test VT     (f) Finetune Both test VT

Figure 7: The first row shows the results of testing the model robustness against horizontal shifts and the second row shows the results against vertical shifts.

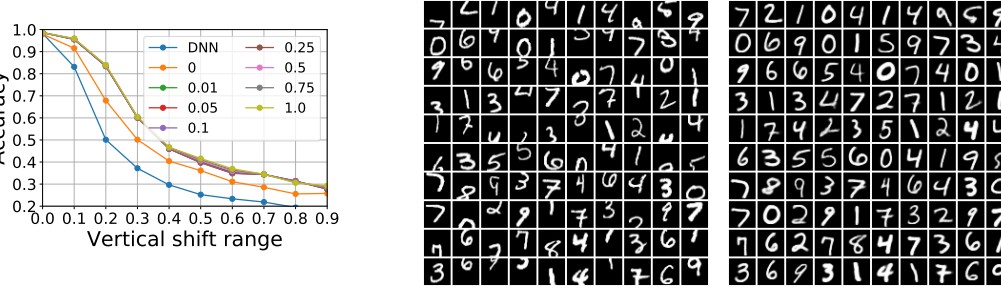

Figure 8: Performance regarding different percentages of test data used for fine-tuning manipulation

(a) Vertically shifted training data     (b) do(m=0) with the $z$ and $y$ from the vertical shifted data

Figure 9: Visualization of the disentangled representation.

**Training with clean MNIST data only.** Figure 7 shows the results for deep CAMA trained on clean data only. Deep CAMA without fine-tuning (orange lines) perform similarly to a DNN (blue lines) on horizontally shifted images, but it is more robust to vertical shifts. The advantage of deep CAMA is clear when fine-tuning is used at test time (green lines): fine-tuning on noisy test data with the same shift clearly improves the robustness of the network (panels 7(b) and 7(d)). We further inspect the generalization of deep CAMA to unseen manipulation after fine-tuning in panels 7(a) and 7(e). The robustness results of fine-tuned models are similar or even slightly better than the models without fine-tuning. This clearly shows that our model is capable of learning manipulations in an unsupervised manner, without deteriorating the generalization ability to unseen manipulations. Lastly, panels 7(c) and 7(f) show the robustness of our model to both shifts when both types of manipulation are used for fine-tuning, and we see clear improvements over both manipulations.

**Training with augmented MNIST data** We explore the setting where the training data is augmented with noisy data. As discussed in Section 3.1, here deep CAMA naturally learns disentangled representation due to its independent mechanism design. Indeed this is confirmed by Figure 9, where panel 9(b) shows the reconstructions of noisy data from panel 9(a) with $do(m = 0)$. In this case the model keeps the identity of the digits but moves them to the center of the image. Recall that $do(m = 0)$ corresponds to clean data which contains centered digits. This shows that deep CAMA can disentangle the intrinsic unknown style $Z$ and the shifting manipulation variable $M$.

We show the robustness results of deep CAMA with augmented training in Figure 10 (cf. Figure 1). Here shift range $0.5$ is used to augment the training data. Take the vertical shift test in panel 10(a) for example. When vertically shifted data are augmented to the training set, the test performance without fine-tuning (green line) is significant better. Further, fine-tuning (brown line) brings in even

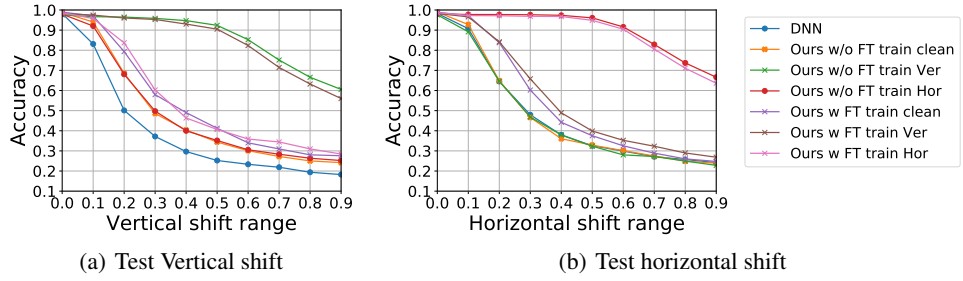

(a) Test Vertical shift      (b) Test horizontal shift

Figure 10: Performance of our model against different manipulation (c.f. Figure 1).

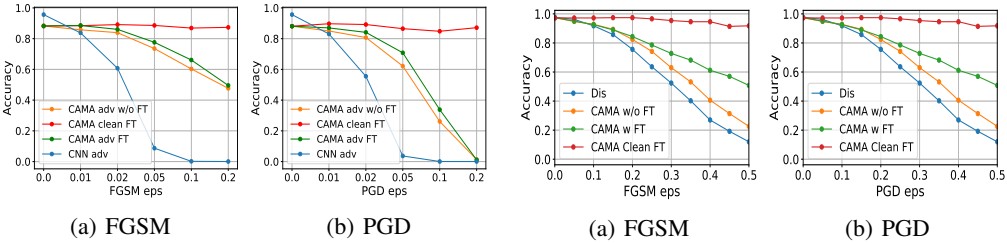

(a) FGSM   (b) PGD     (a) FGSM   (b) PGD

Figure 12: Test accuracy on adversarial examples crafted on CIFAR-binary data.

Figure 13: Test accuracy on adversarial examples crafted on measurement data.

larger improvement for large scale shifts. On the other hand, when using horizontally shifted data in training, deep CAMA's robustness on vertically shifted data also improves (red line), which is different from discriminative DNNs overfitting behaviour (Figure 1). Therefore deep CAMA shows significant advantage over discriminative DNNs as its robustness to unseen manipulations can be improved by observing *other related manipulations*. Our model does not overfit to a specific type of manipulations, at the same time further fine-tuning can always improve the robustness against new manipulations in the test set (pink line). The same conclusion holds in panel 10(b).

We also quantify the amount of noisy data required for fine-tuning in order to improve the robustness of deep CAMA models. As shown in Figure 8, even using $1\%$ of the noisy data is sufficient to learn the vertical shift manipulation presented in the test set.

**Adversarial Attack Test on MNIST**  We further test deep CAMA's robustness against two adversarial attacks: fast gradient sign method (FGSM) (Goodfellow et al., 2014) and projected gradient descent (PGD) (Madry et al., 2017). Note that, these attacks are specially developed for images with the small perturbation constraint. However, theses attack does not have guarantee to be valid by our definition as the manipulation depends on the class label $Y$, which has the risk of changing the ground-truth label. Such risk has also been discussed in Elsayed et al. (2018).

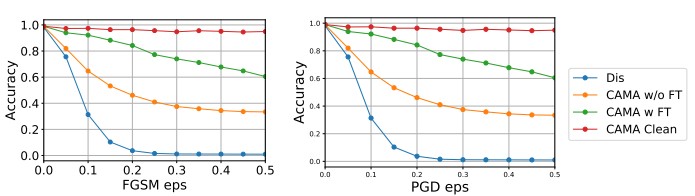

Figure 11: Test accuracy on MNIST adversarial examples.

Figure 11 show the results comparing CAMA and the DNN; both are trained on clean images only. CAMA is significantly more robust to both attacks than DNN (orange line), and with fine-tuning, CAMA shows additional $20\% - 40\%$ accuracy increase. We also show the clean data test accuracy after fine-tuning maintains to be the same thanks to our causal consistent model design.

**Adversarial attack test on natural image classification**  The last experiment in this section evaluates the adversarial robustness of deep CAMA when trained on natural images. In this case we follow Li et al. (2018) and consider *CIFAR-binary*, a binary classification dataset containing airplane and frog images from CIFAR-10. We choose to work with CIFAR-binary because VAE-based

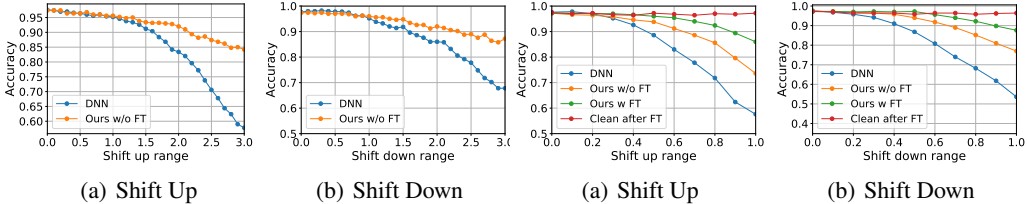

| (a) Shift Up | (b) Shift Down | (a) Shift Up | (b) Shift Down |

Figure 14: Manipulate co-parents              Figure 15: Manipulate children

fully generative classifiers are less satisfactory for classifying clean CIFAR-10 images ($< 50\%$ clean test accuracy). The deep CAMA model trained with data augmentation (adding Gaussian noise with standard deviation 0.1, see objective (5)) achieves $88.85\%$ clean test accuracy on CIFAR-binary, which is on par with the results reported in Li et al. (2018). For reference, a discriminative CNN with $2\times$ more channels achieves $95.60\%$ clean test accuracy. Similar to previous sections we apply FGSM and PGD attacks with different $\epsilon$ values to both deep CAMA and the discriminative CNN, and evaluate classification accuracies on the adversarial examples before and after finetuning.

Results are reported in Figure 12. For both FGSM and PGD tests, we see that deep CAMA, before finetuning, is significantly more robust to adversarial attacks when compared with a discriminative CNN model. Regarding finetuning, although PGD with large distortion ($\epsilon = 0.2$) also fools the finetuning mechanism, in other cases finetuning still provides modest improvements ($5\%$ to $8\%$ when compared with the vanilla deep CAMA model) without deteriorating test accuracy on clean data. Combined with adversarial robustness results on MNIST, we conjecture that with a better generative model on natural images the robustness of deep CAMA can be further improved.

## 4.2 Robustness test on measurement based data with generalized Deep CAMA

Our causal view on valid manipulations allows us to test the robustness of models to generic measurement data. Unfortunately, there exists no public dataset with multiple variables where ground truth causal relationships are known. Therefore we generate synthetic data (see appendix) following a causal process, and test the performance of the generic deep CAMA on this measurement based data. Here we use Gaussian variables for $A$, $C$ and $X$, and categorical variables for $Y$. All the ground truth causal relationships are nonlinear (quadratic mainly).

**Manipulation Test** First, we test manipulations on co-parents, $C$, while keeping the ground truth causal influence from $C$ to $X$ static. Thus, both $C$ and $X$ change. We manipulate $C$ by shifting it up or down, which is a reasonable analogy to the noisiness in measurement data. For example, in medical measurement data, different doctors may have different subjective standards while examining the patients, thus the same measurement can be shifted up or down. Figure 14 shows the result: compared to a discriminatively trained DNN, deep CAMA is significantly more robust to a wide range of manipulations. However, when the range of the shifting manipulations increases, the classification accuracy of the discriminative DNN drops drastically. This confirms our theory in Section 3.2 that manipulations in $C$ do not affect the decision making of deep CAMA, therefore our model is more robust to manipulation on co-parents as compared to discriminative DNNs.

Figure 15 shows the performance of the generic deep CAMA when the children $X$ are manipulated, and the model only sees clean data at training time. While deep CAMA achieves the same accuracy as a discriminative DNN on clean data, it is again significantly more robust to manipulations even without fine-tuning (the orange line vs the blue line). With fine-tuning (green line), the robustness of deep CAMA is further improved, especially when the amount of distortion is large. The red line shows that deep CAMA's test accuracy on clean data, which does not drop after fine-tuning on different shifts. This further confirms that during test time, fine-tuning learns the influence of $M$ without affecting the causal relationships between $Y$ and $Z$.

**Adversarial Attack Test** Lastly we evaluate the adversarial robustness of the generalized CAMA model. We only allow attacks on the children $X$ and coparents $C$ to be consistent with our definition of valid attacks. This applies to both DNN and CAMA. Figure 13 shows the results in terms of test accuracy with adversarial examples generated using FGSM and PGD attack methods. Again deep CAMA demonstrate significantly improved robustness against adversarial attacks, and fine-tuning further provides improvements on robustness while keeping high accuracy on clean test examples.

## 5 RELATED WORK

**Adversarial robustness** Adversarial attacks can easily fool a discriminative DNN for vision/speech/language modelling tasks by adding imperceptible perturbations (Carlini & Wagner, 2018; Alzantot et al., 2018; Carlini & Wagner, 2017b; Szegedy et al., 2013; Papernot et al., 2017). Adversarial training (Madry et al., 2017; Tramèr et al., 2018) has shown some success in defending attacks, however, these techniques assume the knowledge of the adversary and present the perturbation to the model during training. Still, a discriminative model after adversarial training is vulnerable to unseen manipulations. Deep generative modelling has recently been applied as a defence mechanism to adversarial attacks. Specifically, existing work considered de-noising adversarial examples before feeding these inputs to the discriminative classifier (Song et al., 2018; Samangouei et al., 2018). Very recently, research revisited (deep) generative classifiers and provided evidence that they are more robust to adversarial attacks (Li et al., 2018; Schott et al., 2019; Lee et al., 2018).

**Causal learning** Causal inference has a long history in statistical research (Spirtes et al., 2000; Pearl, 2009; Peters et al., 2017; Pearl & Mackenzie, 2018). Although it has fundamental importance, the causal view has not been widely incorporated to the robustness analysis of neural networks on unseen manipulations. The most relevant work is in applying the existing causal views to transfer learning and domain adaption (Zhang et al., 2013; Stojanov et al., 2019; Zhao et al., 2019; Gong et al., 2016), where the difference in various domains are treated as either target shift or conditional shift from a causal perspective. As an extension to the domain adaptation work, Rothenhäusler et al. (2018); Heinze-Deml & Meinshausen (2017); Arjovsky et al. (2019) also discussed learning robust predictors across different domains. However, in these approaches the domain is specified either explicitly or though exemplar paired points, thus an unseen manipulation is not explicitly considered. By contrast, our proposed method does not rely on any given domain information. Another related area is causal feature selection (Aliferis et al., 2010), where causal discovery is applied first and features in the Markov Blanket of the prediction target are selected. We also note that CAMA's design is aligned with causal and anti-causal learning analyses (Schölkopf et al., 2012; Kilbertus et al., 2018), in that CAMA models the causal mechanism $Y \rightarrow X$ and use Bayes' rule for anti-causal prediction. Different from Schölkopf et al. (2012), CAMA is not limited to only two endogenous variables; rather it provides more generic design handling latent causes that correspond to both intrinsic variations and data manipulations.

**Disentangled representations** Learning disentangled representations has become a hot topic of research in recent deep generative modelling literature. A considerable amount of effort went to developing training objectives for variational auto-encoders, e.g. $\beta$-VAE (Higgins et al., 2017) and other information theoretic approaches (Kim & Mnih, 2018; Chen et al., 2018). Additionally, different factorization structure in graphical model design has also been explored for disentanglement (Narayanaswamy et al., 2017; Li & Mandt, 2018).

## 6 DISCUSSION

We have provided a causal view on the robustness of neural networks, showing that the vulnerability of discriminative DNNs is due to the lack of causal reasoning. We defined valid manipulations under this causal view, which are the manipulations on the children and/or the co-parents of the target variables, independent of the target and/or the cause of the target. We further proposed a deep causal manipulation augmented model (deep CAMA), which follows the causal relationship in the model design, and can be adapted to unseen manipulations at test time. Our model has demonstrated improved robustness, even without adversarial training. When manipulated data are available, our model's robustness increases for both seen and unseen manipulation.

Our framework is generic, however, manipulations can change over time, and a robust model should adapt to these perturbations in a continuous manner. Our framework thus should be adapted to online learning or continual learning settings. In future work, we will explore the continual learning setting of deep CAMA where new manipulations come in a sequence. In addition, our method is designed for generic class-independent manipulations, therefore a natual extension would consider class-dependent manipulations where $M$ is an effect of $Y$. Lastly out design excludes gradient-based adversarial attacks which is dependent on both the target and the victim model. As such attacks are commonly adopted in machine learning, we would also like to extend our model to such scenarios.

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

# A  DERIVATION DETAILS

## A.1  THE INTERVENTION ELBO

When training with clean data $\mathcal{D} = \{(x_n, y_n)\}$, we set the manipulation variable $M$ to a null value, e.g. $do(m = 0)$. In this case we would like to maximise the log-likelihood of the *intervened* model, i.e.

$$\max_{\theta} \mathbb{E}_{\mathcal{D}}[\log p_{\theta}(x, y | do(m = 0))].$$

This log-likelihood of the intervened model is defined by integrating out the unobserved latent variable $Z$ in the intervened joint distribution, and from do-calculus we have

$$\log p_{\theta}(x, y | do(m = 0)) = \log \int p_{\theta}(x, y, z | do(m = 0)) dz$$
$$= \log \int p_{\theta}(x | y, z, m = 0) p(y) p(z) dz. \tag{12}$$

A variational lower-bound (or ELBO) of the log-likelihood uses a variational distribution $q(z|\cdot)$

$$\log p_{\theta}(x, y | do(m = 0)) = \log \int p_{\theta}(x | y, z, m = 0) p(y) p(z) \frac{q(z|\cdot)}{q(z|\cdot)} dz$$
$$\geq \mathbb{E}_{q(z|\cdot)} \left[ \log \frac{p_{\theta}(x | y, z, m = 0) p(y) p(z)}{q(z|\cdot)} \right]. \tag{13}$$

The lower-bound holds for arbitrary $q(z|\cdot)$ as long as it is absolutely continuous w.r.t. the posterior distribution $p_{\theta}(z|x, y, do(m = 0))$ of the intervened model. Now recall the design of the inference network/variational distribution in the main text:

$$q_{\phi}(z, m | x, y) = q_{\phi_1}(z | x, y, m) q_{\phi_2}(m | x),$$

where $\phi_1$ and $\phi_2$ are the inference network parameters of the corresponding variational distributions. Performing an intervention $do(m = 0)$ on this $q$ distribution gives

$$q_{\phi}(z | x, y, do(m = 0)) = q_{\phi_1}(z | x, y, m = 0).$$

Defining $q(z|\cdot) = q_{\phi_1}(z | x, y, do(m = 0))$ and plugging-in it to eq. (13) return the *intervention* ELBO objective (3) presented in the main text.

## A.2  THE ELBO FOR UNLABELLED TEST DATA

The proposed fine-tuning method in the main text require optimising the marginal log-likelihood $\log p_{\theta}(x)$ for $x \sim \tilde{\mathcal{D}}$, which is clearly intractable. Instead of using a variational distribution for the unobserved class label $Y$, we consider the variational lower-bound of $\log p_{\theta}(x, y)$ for all possible $y = y_c$:

$$\log p_{\theta}(x, y) = \log \int p_{\theta}(x, y, z, m) dz dm$$
$$= \log \int p_{\theta}(x, y, z, m) \frac{q_{\phi}(z, m | x, y)}{q_{\phi}(z, m | x, y)} dz dm \tag{14}$$
$$\geq \mathbb{E}_{q_{\phi}(z, m | x, y)} \left[ \log \frac{p_{\theta}(x, y, z, m)}{q_{\phi}(z, m | x, y)} \right] := \text{ELBO}(x, y).$$

Since both logarithm and exponent functions preserve monotonicity, and for all $y_c, c = 1, ..., C$ we have $\log p_{\theta}(x, y_c) \geq \text{ELBO}(x, y_c)$, we have

$$\log p_{\theta}(x, y_c) \geq \text{ELBO}(x, y_c), \forall c \implies p_{\theta}(x, y_c) \geq \exp[\text{ELBO}(x, y_c)], \forall c$$

$$\implies \log p(x) = \log \left[ \sum_{c=1}^{C} p_{\theta}(x, y_c) \right] \geq \log \left[ \sum_{c=1}^{C} \exp[\text{ELBO}(x, y_c)] \right] := \text{ELBO}(x),$$

which justifies the ELBO objective (7) defined in the main text.

# B ADDITIONAL RESULTS

**CNN** We also performed experiments using different DNN network architectures. The convolution layers in CNN are designed to be robust to shifts. Thus, we test these vertical and horizontal shifts with a standard CNN architecture as used in `https://keras.io/examples/cifar10_cnn/`. 4 convolution layers are used in this architecture.

Figure 16 shows the performance against different shifts. We see that adding vertical shifts to the training data clearly harmed the robustness performances to unseen horizontal shifts as shown in 17(b). Adding horizontal shifted images in training did not influences the performance on vertical shifts much. Thus, we see that using different architectures of DNN, even the one that are designed to be robust to these manipulations, lack of generalization ability to unseen data is a common problem.

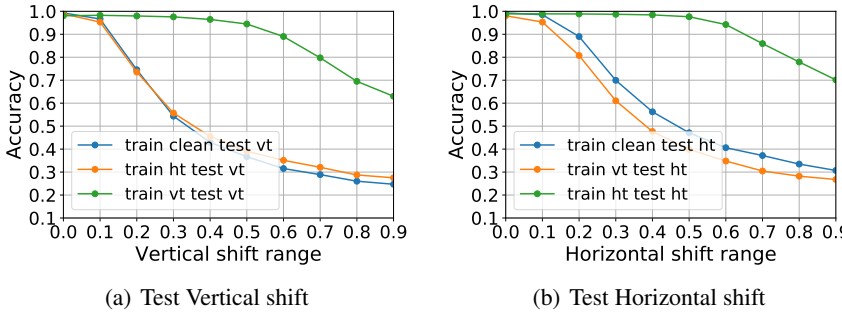

(a) Test Vertical shift    (b) Test Horizontal shift

Figure 16: Robustness results for DNNs against different manipulations on MNIST using CNN. Panels (a) and (b) show the accuracy on classifying noisy test data generated by shifting the digits vertically (vt) and horizontally (ht). It shows that data augmentation during training makes generalization to unseen shifts worse (orange versus blue lines).

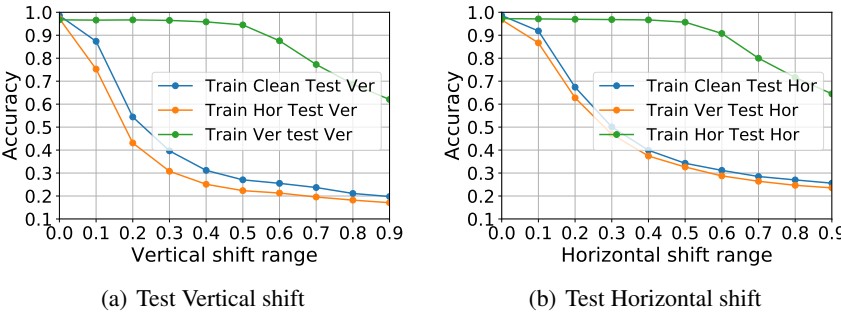

(a) Test Vertical shift    (b) Test Horizontal shift

Figure 17: Robustness results for DNNs against different manipulations on MNIST using a large MLP. Panels (a) and (b) show the accuracy on classifying noisy test data generated by shifting the digits vertically (vt) and horizontally (ht). It shows that data augmentation during training makes generalization to unseen shifts worse (orange versus blue lines).

**Enlarge Network Size** Here we exam whether network capacity has any influence on the robustness performance to unseen manipulation. We use a wider network with [1024, 512, 512, 1024] units in each hidden layer instead of [512, 256, 126, 512] sized network in the paper. Figure 17 shows the robustness performance using this enlarged network. We observe the similar degree of over-fitting to the augmented data. The penalization ability shows no improvement by enlarging the network sizes.

**ZCA Whitening Manipulation** Our result does not limited to shifts, it generalizes to other manipulations. Figure 18 compare the result from training with clean images and training with ZCA whitening images added. We see that adding ZCA whitening images in training harm both robustness against vertical shift and horizontal shift.

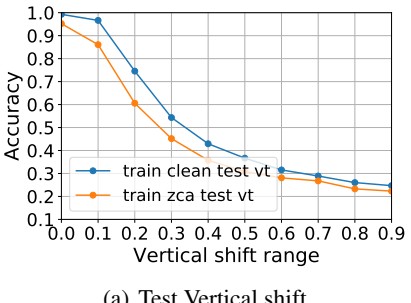
(a) Test Vertical shift

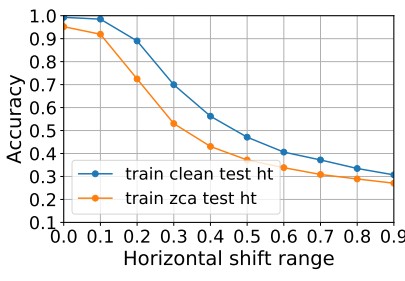
(b) Test Horizontal shift

Figure 18: ZCA Whitening manipulation result. Figure shows the robustness results for DNNs against different manipulations on MNIST using CNN. The blue curve shows that result from training with clean data. The orange curve shows that result from training with zca whitening data added.

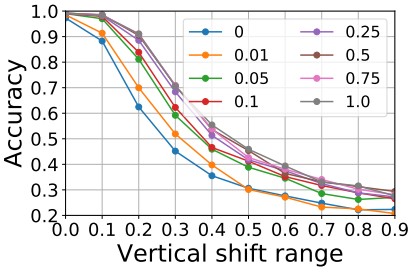

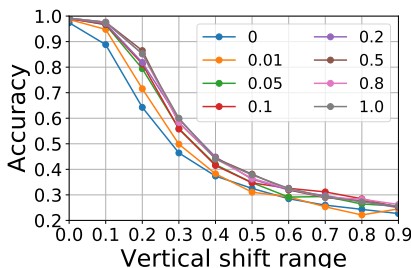

Figure 19: Performance regarding different percentage of test data used for fine-tuning manipulation of horizontal shift without using $do(m) = 0$ for the cleaning training data during fine-tuning.

Figure 20: Performance regarding different percentage of test data used for fine-tuning manipulation of vertical shift using $do(m) = 0$ for the cleaning training data during fine-tuning.

**Additional Figures** In addition to Figure 8, We also show the result testing with Vertical shift show in Figure 19, where a smaller $N_M^p$ network ([dimM, 500, 500]) is used. The conclusion is the same was using the vertical shift. We need very few data for fine-tune. More than $1\%$ data is sufficient.

Similar as Figure 8, we show the result using different percentage of data for fine-tuning in this experiment setting in 20.

## C   EXPERIMENTAL SETTINGS

**Network architecture**

- MNIST experiments:
    - Discriminative DNN: The discriminate model used in the paper contains 4 densely connected hidden layer of $[512, 256, 126, 512]$ width for each layer. ReLU activations and dropout are used with dropout rate $[0.25, 0.25, 0.25, 0.5]$ for each layer.
    - Deep CAMA's $p$ networks: we use $\dim(Y) = 10, \dim(Z) = 64$ and $\dim(M) = 32$.
      $\text{NN}_Y^p$: an MLP of layer sizes $[\dim(Y), 500, 500]$ and ReLU activations.
      $\text{NN}_Z^p$: an MLP of layer sizes $[\dim(Z), 500, 500]$ and ReLU activations.
      $\text{NN}_M^p$: an MLP of layer sizes $[\dim(M), 500, 500, 500, 500]$ and ReLU activations.
      $\text{NN}_{\text{merge}}^p$: an projection layer which projects the feature outputs from the previous networks to a 3D tensor of shape $(4, 4, 64)$, followed by 3 deconvolutional layers with stride 2, SAME padding, filter size $(3, 3, 64, 64)$ except for the last layer $(3, 3, 64, 1)$.

All the layers use ReLU activations except for the last layer, which uses sigmoid activation.

- Deep CAMA's $q$ networks:

  $NN_M^q$: it starts from a convolutional neural network (CNN) with 3 blocks of $\{conv3 \times 3, max\text{-}pool\}$ layers with output channel size 64, stride 1 and SAME padding, then performs a reshape-to-vector operation and transforms this vector with an MLP of layer sizes $[4 \times 4 \times 64, 500, \dim(M) \times 2]$ to generate the mean and log-variance of $q(m|x)$. All the layers use ReLU activation except for the last layer, which uses linear activation.

  $NN_Z^q$: first it uses a CNN with similar architecture as $NN_q^M$'s CNN (except that the filter size is 5) to process $x$. Then after the reshape-to-vector operation, the vector first gets transformed by an MLP of size $[4 \times 4 \times 64, 500]$, then it gets combined with $y$ and $m$ and passed through another MLP of size $[500+\dim(Y)+\dim(M), 500, \dim(Z) \times 2]$ to obtain the mean and log-variance of $q(z|x, y, m)$. All the layers use ReLU activation except for the last layer, which uses linear activation.

- Measurement data experiments:

  - Discriminative DNN: The $A, C, X$ variables are concatenated to an input vector of total dimension 20. Then the DNN contains 3 densely connected hidden layer of $[64, 16, 32]$ width for each layer, and output $Y$. ReLU activations and dropout are used with dropout rate $[0.25, 0.25, 0.5]$ for each layer.

  - Deep CAMA's $p$ networks: we use $\dim(Y) = 5, \dim(A) = 5, \dim(C) = 5, \dim(Z) = 64$ and $\dim(M) = 32$.

    $p(y|a)$: an MLP of layer sizes $[\dim(A), 500, 500, \dim(Y)]$, ReLU activations except for the last layer (softmax).

    $p(x|y, c, z, m)$ contains 5 networks: 4 networks $\{NN_Y^p, NN_C^p, NN_Z^p, NN_M^p\}$ to process each of the parents of $X$, followed by a merging network.

    $NN_Y^p$: an MLP of layer sizes $[\dim(Y), 500, 500]$ and ReLU activations.

    $NN_C^p$: an MLP of layer sizes $[\dim(C), 500, 500]$ and ReLU activations.

    $NN_Z^p$ an MLP of layer sizes $[\dim(Z), 500, 500]$ and ReLU activations.

    $NN_M^p$: an MLP of layer sizes $[\dim(M), 500, 500, 500, 500]$ and ReLU activations.

    $NN_{merge}^p$: it first start from a concatenation of the feature outputs from the above 4 networks, then transforms the concatenated vector with an MLP of layer sizes $[500 \times 4, 500, \dim(X)]$ to output the mean of $x$. All the layers use ReLU activations except for the last layer, which uses linear activation.

  - Deep CAMA's $q$ networks:

    $q(m|x)$: it uses an MLP of layer sizes $[\dim(X), 500, 500, \dim(M) \times 2]$ to obtain the mean and log-variance. All the layers use ReLU activations except for the last layer, which uses linear activation.

    $q(z|x, y, m, a, c)$: it first concatenates $x, y, m, a, c$ into a vecto, then uses an MLP of layer sizes $[\dim(X) + \dim(Y) + \dim(M) + \dim(A) + \dim(C), 500, 500, \dim(Z) \times 2]$ to transform this vector into the mean and log-variance of $q(z|x, y, m, a, c)$. All the layers use ReLU activations except for the last layer, which uses linear activation.

- CIFAR-binary experiments:

  - Discriminative CNN: The discriminate model used in the paper is a CNN with 3 convolutional layers of filter width 3 and channel sizes [128, 128, 128], followed by a flattening operation and a 2-hidden layer MLP of size $[4 \times 4 \times 128, 1000, 1000, 10]$. It uses ReLU activations and max pooling for the convolutional layers.

  - Deep CAMA's $p$ networks: we use $\dim(Y) = 10, \dim(Z) = 128$ and $\dim(M) = 64$.

    $NN_Y^p$: an MLP of layer sizes $[\dim(Y), 1000, 1000]$ and ReLU activations.

    $NN_Z^p$: an MLP of layer sizes $[\dim(Z), 1000, 1000]$ and ReLU activations.

    $NN_M^p$: an MLP of layer sizes $[\dim(M), 1000, 1000, 1000]$ and ReLU activations.

    $NN_{merge}^p$: an projection layer which projects the feature outputs from the previous networks to a 3D tensor of shape $(4, 4, 64)$, followed by 4 deconvolutional layers with stride 2, SAME padding, filter size $(3, 3, 64, 64)$ except for the last layer $(3, 3, 64, 3)$. All the layers use ReLU activations except for the last layer, which uses sigmoid activation.

– Deep CAMA's $q$ networks:

$\text{NN}_M^q$: it starts from a convolutional neural network (CNN) with 3 blocks of $\{\text{conv3} \times 3, \text{max-pool}\}$ layers with output channel size 64, stride 1 and SAME padding, then performs a reshape-to-vector operation and transforms this vector with an MLP of layer sizes $[4 \times 4 \times 64, 1000, 1000, \dim(M) \times 2]$ to generate the mean and log-variance of $q(m|x)$. All the layers use ReLU activation except for the last layer, which uses linear activation.

$\text{NN}_Z^q$: first it re-uses $\text{NN}_M^q$ CNN network for feature extraction on $x$. Then after the reshape-to-vector operation, the vector gets combined with $y$ and $m$ and passed through another MLP of size $[4 \times 4 \times 64 + \dim(Y) + \dim(M), 1000, 1000, \dim(Z) \times 2]$ to obtain the mean and log-variance of $q(z|x, y, m)$. All the layers use ReLU activation except for the last layer, which uses linear activation.

**Measurement data generation**     We set the target $Y$ to be categorical, its children, co-parents and parents are continuous variables. The set 5 classes for $Y$, and $Y$ has 10 children variables and 5 co-parents variables, also one 5 dimensional parents.

Parents ($A$) and co-parents ($C$) are generated by sampling from a normal distribution. We generate $Y$ using structured equation $Y = f_y(A) + \sigma_Y$. We use $f_y = \text{argmax } g(A)$ and $g()$ is a quadratic function $0.2 * A^2 - 0.8A$. $\sigma_Y$ is the Gaussain noise.

To generate the children $X = f(Y, C) + \sigma_x$, we also used quadratic function $f$ and the parameters were sampled from a Gaussian distribution. As in the experiment, we were using fixed scale shift, we also added a normalize the children before adding the Gaussian random noise $\sigma_x$. So that all observations are in similar scale.

**Other**     For MNIST experiments, we uses $5\%$ of the training data as the validation set. We used the training results with the highest validation accuracy for testing. If not otherwise specified, $50\%$ of noisy test data are used for fine-tuning in the shift experiments and all data are used for fine-tuning in the attack experiments.

For the experiments with measurement data. We generated 1000 data in total. We split, 500 data for testing, 450 for training and 50 for validation. We used the training results with the highest validation accuracy for testing for both deep CAMA and for DNN.

