# OpenReview forum: "A Causal View on Robustness  of Neural Networks"
_ICLR.cc/2020/Conference — Reject_

### Official Review · AnonReviewer1 · 2019-10-17
**Official Blind Review #1**

**Rating:** 8

**Review:**

The paper tackles a very important problem, manipulation robustness of modern machine learning models, by applying a chain of design choices perfectly well:
  * VAEs as probabilistic generators
  * Manipulations as independent causes that can be generated by interventions
  * Causal inference for manipulation-corrected training
  * Bayesian inference for robust prediction

Sticking to the Bayesian point of view, the method can perform model averaging across potential manipulations at the test time. This is an extremely elegant property, which is also proven in the experiments to be very effective.

Being able to learn previously unseen types of manipulations only by proper application of causal inference tools is a very important news for the adversarial robustness community.

Figure 9 is a spectacular proof of concept to illustrate the disentanglement property of the proposed method.

I have only one point for improvement. I do not buy the argument that we should do q(m|x) but not q(m|x,y). Why should we exclude class-specific manipulations? This wouldn't affect the cause (the class) but only the outcome, so would be a valid manipulation. I would actually expect the model to work still well with q(m|x,y). Could the authors comment on what the concrete benefit of leaving out y from manipulations is?

The construction of the loss in Eq 5 is in spirit semi-supervised learning on m. We show the true m=0 cases to the model during training but assume not to know the labels of the manipulated samples. It could be beneficial to draw a link to semi-supervised learning here and even tie it to the arguments about the relationship between causal inference and semi-supervised learning:

Schölkopf et all., On Causal and Anticausal Learning, ICML, 2012.


**Experience Assessment:**

I have read many papers in this area.

**Review Assessment: Checking Correctness Of Derivations And Theory:**

I carefully checked the derivations and theory.

**Review Assessment: Checking Correctness Of Experiments:**

I assessed the sensibility of the experiments.

**Review Assessment: Thoroughness In Paper Reading:**

I read the paper thoroughly.

---

> ### Author Response · Authors · 2019-11-15
> **Response to Reviewer #1**
>
> We thank the reviewer for valuable suggestions.
>
> > class-specific manipulation and $q(m| x, y)$.
>
>  From the description of the question, it seems to us that the reviewer was asking about the generative model design and suggesting class-dependant manipulations.
> We agree with the reviewer that class-dependant manipulation is a valid setting for some applications, for example, only small stickers can be put on traffic signs while big paintings can be put on a wall.  However, we believe that class-independent manipulation is more general and common in typical applications such as image classification. Manipulations such as rotation, shifting, camera noise, and compression artifacts are very common in natural images, all of them are independent of the image classes.
>
> Moreover, extending CAMA to the class-dependent manipulation case is straight-forward. In this case, we can add the Y->M connection to the causal graph and modify the probabilistic generative model accordingly. This means our model will be
> $p( x, y, z, m) = p(y)p(z)p(m|y)p(x|y,z,m)$, and the only difference is to use $p(m|y)$ instead of $p(m)$ in the generative model. Training, fine-tuning and prediction for such a modified model still follow the proposed methods in the main text, therefore we see this modification as a straight-forward extension. We will add such a new model in an extended version of the work as we added in the discussion part of the revised version of the paper.
>
> Regarding the designed choice of using $q(m|x)$ instead of $q(m | x, y)$:
>
> $q(m|x)$ is the inference network for $M$ which contributes to inferring the posterior of the latent variables. This design is consistent with the causal graph stating that $M$ and $Y$ are independent causes of $X$. If class-dependant manipulation is in consideration, the generative model $p(x, y, z, m)$ will be modified to reflect this causal relationship, and in this case, $q(m | x, y)$ will be the preferred design choice for the inference network.
>
> Additionally, using $q(m|x)$ is more computationally efficient since in test time label $Y$ is unavailable. Certainly using $q(m | x, y)$ and iterate over all possible classes $y = y_c, c = 1, …, C$ is possible, but it also brings in more computational burden.
>
>
> > Connection to Schölkopf et al 2012
>
> We thank the reviewer for making this connection.
> Indeed, deep CAMA algorithm is related to [1] where the causal and anti-causal learning are discussed. Deep CAMA's designs of modeling the causal direction $Y \rightarrow X$ and using Bayes' rule for anti-causal prediction are well justified by the analyses in [1].
> Note that, [1] mainly discussed the setting with only one cause and one effect variable (two endogenous variables in total), with independent noise treated as exogenous variables.  If only considering either the $(M, X)$ pair or the $(Y, X)$ pair, it is connected to [1]’s analysis on semi-supervised learning and robustness analysis in general. As suggested, our design for the $do(m)$ procedure also allows the “labeling” on $M$ which follows the semi-supervised setting.
> However, our model deals with a more general setting, which allows both observed cause $Y$ and latent causes $Z, M$.
>
> We have added two citations [1] and [2] and some brief discussions regarding these works in the related work section. We do appreciate other insights beyond robustness that [1] and [2] provided, but it is out of the scope of our work.
>
> [1] Schölkopf et all., On Causal and Anticausal Learning, ICML, 2012.
> [2] Kilbertus, N., Parascandolo, G., & Schölkopf, B. (2018). Generalization in anti-causal learning. arXiv preprint arXiv:1812.00524.

---

### Official Review · AnonReviewer2 · 2019-10-27
**Official Blind Review #2**

**Rating:** 8

**Review:**

In this paper, the author first assume the data generation process that sample X is generated by label Y, latent style (domain information) Z and other manipulations M and then propose the deep causal manipulation augmented model that use the do calculus to model the manipulations of data and further take it as the cause to the observed effect variables x. The author further elaborately devises experiment on the standard dataset and achieve good results.

Strong point:
(1)	The paper is well-written and the idea is motivated.
(2)	The experiment is convincing and the result is good.

Weak point:
(1)	Please provide the deducing process of the ELBO, such as how the ELBO is deduced in the training mode and the prediction process should be elaborated detailedly.
(2)	The symbolism is confusing, such as the \phi_1 and \phi_2 in Eq(6).
(3)	What is the difference between manipulations M and the latent style Z? Do they belong to the domain information? Please justify.


**Experience Assessment:**

I have published in this field for several years.

**Review Assessment: Checking Correctness Of Derivations And Theory:**

I assessed the sensibility of the derivations and theory.

**Review Assessment: Checking Correctness Of Experiments:**

I assessed the sensibility of the experiments.

**Review Assessment: Thoroughness In Paper Reading:**

I read the paper at least twice and used my best judgement in assessing the paper.

---

> ### Author Response · Authors · 2019-11-15
> **Response to Reviewer #2**
>
> We thank the reviewer for constructive feedback. We have addressed your concern as follows and updated the paper accordingly:
>
> > Deducing process of the ELBO
>
> We have added the complete derivation of the ELBO objectives in appendix A in the revised version of the paper.
>
> > The symbolism
>
> We have added text explanation after Eq (2) for $\phi_1$ and $\phi_2$ to improve the clarity of the symbols.
> We use $\phi$ to denote all the parameters of the encoder network and $\phi = \{ \phi_1, \phi_2 \}$, where $\phi_1$ is the parameter for the encoder network for the variational distribution $q(z | x, y, m)$, and $\phi_2$ is used for the $q(m | x)$ part.
> With respect to Figure 4, $\phi_1$ is the parameter for $NN_Z^q$ network and $\phi_2$ is the parameter for $NN_M^q$ network
>
> > Difference between $M$ and $Z$
>
> $M$ reflects the manipulation that we wish the model to be robust to. These manipulations are the variations that clean data naturally do not contain, and they may be unseen in the training set which reflects the real-world application need.
> $Z$ models the natural variations for data from a domain, and it remains to be the same in all manipulated data.
>
> Using shifted MNIST data as an example: $Z$ is learned to represent factors related to writing styles, e.g. thickness of the stroke, while $M$ represents the performed shift on the observed data, e.g. horizontal/vertical shift and the specific shift range applied to the clean example. Indeed Figure 9 clearly shows the disentanglement of $Z$ & $M$ representations, in the sense that reconstruction with intervention $do(m=0)$ shifts the digits back to the center.

---

### Official Review · AnonReviewer4 · 2019-11-15
**Official Blind Review #4**

**Rating:** 3

**Review:**

Summary:
(1) The paper makes vague and unsupported claims about causality.
   The notion of 'causal', as well as do() notation, is really hollowed out here. It is of limited significance. In Equation (3) do() disappears from one line to the next, illustrating how little it does.

(2) Lack of novelty: The paper references "Learning disentangled representations with Semi-supervised deep generative models", but fails to mention that their model is is nearly identical.

(3) The quality is on the low side; confusing mistakes in figures, in some
 sections poor proof-reading.

Details:
(1) Exaggerated claims.
"We argue that the incapability for causal reasoning is the reason of DNN's vulnerability".  (This is stated twice in different words).

  Unfortunately, the term "causal reasoning" is used without making it precise.  The paper has a very weak relation to causality; one could replace "causal" with "translation invariant", and very little would change.  Their experiments try to learn translation invariance (appendix shows whitening invariance).

The paper makes gratuitous use of do() notation; for example, regular MNIST is reinterpreted as not being observational, but as arising from an intervention, namely do(translation=0).  But do() does not amount to much here. It reduces to mere conditioning here, as the do is always applied to a root node without parents.  This is clear in equation in 3, where do's are correctly just deleted from one line to the next; do(), and 'causal' are used mostly as a rhetorical device.

(2) The model in the paper is pretty much the same as "Learning disentangled representations with Semi-supervised deep generative models", figure 2 here corresponds to figure 4 there.  Both papers use partially supervised (or unsupervised) VAEs. There are some differences, e.g. the prior work considers a more general case and uses importance sampling.  This paper refers but fails to say how similar they are.  There should be a comparison between the difference inference methods used.


(3) The paper is sloppy in places.

Figure 6: the dotted arrow between nodes A and Z should not be there.
It does not follow from the graphical model in Figure 5.

In Figure 12a),  the text contradicts the figure. The brown curve with fine tuning has lower accuracy than the green one without fine tuning, but the text says the opposite.

Figure 11: there is a curve mislabelled as "Dis"

Results are mixed.  While results on vertical translation invariance and adversarial robustness seem good, the method fails to achieve horizontal translation invariance (Figure 7 b) and c)) - disappointing. But some of the translation invariance seems to be due to just using a generative model (the VAE), as it occurs without any fine-tuning.


**Experience Assessment:**

I have read many papers in this area.

**Review Assessment: Checking Correctness Of Derivations And Theory:**

I assessed the sensibility of the derivations and theory.

**Review Assessment: Checking Correctness Of Experiments:**

I assessed the sensibility of the experiments.

**Review Assessment: Thoroughness In Paper Reading:**

I read the paper at least twice and used my best judgement in assessing the paper.

---

### Author Response · Authors · 2020-06-19
**reply to reviewer 4**

We would like to add a reply to reviewer 4 now to clarify the misunderstanding as the review was in just after the rebuttal deadline making us unable to reply during the reviewing process. We would also like to thank the other two reviewer’s positive reviews again.

> ...claims about causality not needed..  Unfortunately, the term "causal reasoning" is used without making it precise.  The paper has a very weak relation to causality; one could replace "causal" with "translation invariant", and very little would change.

We agree that one can replace ``causality” with another set of words describing its properties, but here by “causal” we mean much more than “translation invariant.” The causal perspective is crucial to both the definition of valid data manipulations and the construction of the proposed CAMA model and implies the modularity properties of the model..

In this work, we assume that the causal relationship among the observed variables is given. Specifically, we require the specification of the observed variables as either an ancestor ($A$), or a co-parent ($C$) or a child ($X$) of the target variable $Y$. Note that changes in the parents of $Y$ (including $A$) will cause changes in $Y$, so without the specification of a causal relationship, simply perturbing all the observed variables and maintaining the same label $Y$ does not correspond to an invalid manipulation. From this perspective, the causal view is important for the definition of valid manipulations.

The deep CAMA model is inspired by the causal process and relies upon its properties. Specifically, in the generic measurement data case, it requires a causally consistent specification of the $C$, $X$, and $A$ variables in the graphical model. Replacing a “causal relationship” with “translation-invariant” cannot fully describe the exploited properties of the causal graph structure. From this perspective, the causal view is important for the construction of our model.


> In Equation (3) do() disappears from one line to the next. Do operation is not needed

In the case of single modality data, our model design is applicable to both situations where $M$ is the root node or non-root node in the causal graph. To see this, we consider two possible scenarios for the causal graph. First, when $M$ is a root node in the causal graph, the do() operation reduces to conditioning. Second and more interestingly, when $M$ is not a root node (e.g. there might exist causal relationships between $Y$ and $M$), applying the do() operation will intervene on $M$, which cuts off the influence from $M$’s parents (in the example the influence from $Y$), and the generation process of manipulated data reduces to the root node $M$ case. Therefore this do() operation enables the deep CAMA model design to handle manipulated data coming from both cases.



(2) Lack of novelty: similarity to  "Learning disentangled representations with Semi-supervised deep generative models"(Narayanaswamy et.al)

Narayanaswamy et.al. proposed a semi-supervised learning algorithm to learn disentangled representation for computer vision tasks. Their approach extends VAE with two latent variables $Y$ and $Z$ to model images $X$. They provide partially observed $Y$ which are ``interpretable''  depending on the computer vision application context. They achieve meaningful synthetic image generation by sample different latent variables in $Y$ and $Z$.

In contrast, our work considers model robustness from a causal perspective, and build models informed by our causal analysis. To the best of our knowledge, none of the following aspects are considered in Narayanaswamy et.al:

The proposed deep CAMA model focuses on the robustness of neural networks to **unseen** manipulations, which benefits from a causal view. First, the latent variables have very different meanings. Apart from the fact that $Y$ is solely used to represent the prediction target, the $M$ and $Z$ variables are designed to separate the latent factors that can or cannot be **artificially** intervened by the adversary.
The fine-tuning algorithm provides a **test-time** adaptation scheme for the deep CAMA model (thus enabling adaptation to any unseen manipulations). Importantly, the fine-tuning method updates the model parameter in a selective manner, motivated by the modularity property of the causal generation process of noisy data.
Deep CAMA applies to generic measurement data which is beyond computer vision tasks. In this case, the causal graph of the data generation process plays a key role in model design, since now deep CAMA also models all the variables in the Markov blanket of $Y$, which is clearly beyond the factor model considered in Narayanaswamy et.al. Our empirical study clearly shows that a consistent design of the model to the underlying causal graph is key to both the robustness of the model and the efficiency of fine-tuning for unseen manipulations.

We have revised our paper to further clarify these.

---

### Decision · Program_Chairs · 2019-12-19

**Decision:**

Reject

**Comment:**

This paper attempts to present a causal view of robustness in classifiers, which is a very important area of research.
However, the connection to causality with the presented model is very thin and, in fact, mathematically unnecessary. Interventions are only applied to root nodes (as pointed out by R4) so they just amount to standard conditioning on the variable "M". The experimental results could be obtained without any mention to causal interventions.